# Standing Steadiness and Asymmetry after High Tibial Osteotomy Surgery: A 2 Year Follow-Up Study

**DOI:** 10.3390/jpm12101594

**Published:** 2022-09-27

**Authors:** Eduard Kurz, Kay Brehme, Thomas Bartels, Martin Pyschik, Manuel Jenz, Wiebke Kadler, Karl-Stefan Delank, René Schwesig

**Affiliations:** 1Department of Orthopedic and Trauma Surgery, Faculty of Medicine, Martin-Luther University Halle-Wittenberg, 06120 Halle (Saale), Germany; 2Sports Clinic Halle, Center of Joint Surgery, 06108 Halle (Saale), Germany

**Keywords:** knee osteoarthrosis, surgery, posturography, pain, quality of life, long term effects

## Abstract

(1) Background: Knee osteoarthritis (OA) is a serious orthopedic problem. In this context, the high tibial osteotomy (HTO) is an established surgical procedure to reduce the load and degeneration of the affected compartment. The aim of this investigation was to judge standing steadiness and asymmetry, pain intensity and quality of life among patients who underwent HTO surgery. (2) Methods: Twenty-five male patients with medial tibiofemoral OA finished this 2 year follow-up study. Standing balance was captured using force plates with four uniaxial sensors before, 6 weeks after, 1 year after, and 2 years after HTO surgery. The percentage weight (PW) under the foot at one side, the stability (ST) index and the weight distribution (WD) index were the main outcomes. Comparisons were conducted using repeated measures analyses of variance. (3) Results: Over time, the PW under the foot at the HTO side increased on average (*p* < 0.001). In terms of standing steadiness, the average ST remained similar over the time points (*p* = 0.71). The WD index was affected by time (*p* = 0.003). (4) Conclusions: In order to judge short-term effects, the PW is recommended, whereas long-term effects can be identified either through the PW or the WD index.

## 1. Introduction

Shifts in the load axis of the lower extremities in the sense of a varus or valgus position and the associated increased unicompartmental pressure load can contribute to the development and progression of early knee osteoarthritis (OA) with chronic pain. In addition to surgical interventions such as total or partial joint replacement surgeries, osteotomies close to the knee joint in particular offer a causal treatment approach for patients with frontal plane malalignment of the lower limb [1]. The aim of high tibial osteotomy (HTO) surgery is to provide relief to the overloaded knee compartment and restore the physiological limb alignment. Patients with an excessive genu varum suffer from on average a two-fold increased risk of knee osteoarthritis, accompanying an approximately three-fold increased disease progression compared to a sample with a normal lower limb alignment [2]. It is well known that patients with medial knee osteoarthritis suffer from functional limitations in everyday life due to stress-related pain, depending on the severity of the symptoms. As a result of a leg alignment surgery, the majority have significantly benefited from an increase in quality of life [3,4].

So far, there are only a few studies that deal with the postoperative effects of HTO surgery on postural control [5,6]. Hunt et al. [5] observed 1 year postoperatively that single-leg steadiness in patients with knee osteoarthritis is not significantly different from that prior to HTO surgery. Although not significant, reported effects were up to 0.33 and thus above trivial. Standing on one leg is challenging for patients and older participants [5,7,8]. Therefore, only 10 s trials were conducted. Possibly, longer standing duration times would have increased the effects measured. For assessments of standing steadiness, segments of at least 30 seconds’ duration are recommended [9]. Posturographic systems with more than one force plate in particular overcome these limitations [10,11]. Bipedal assessments enable longer conduction times, while multiple force plates make it possible to estimate standing asymmetry (weight distribution).

In contrast, long-term follow-ups employing any kind of posturography in order to judge standing balance and weight distribution are not yet available. Thus, the main intentions of this study were to evaluate the influence of a surgical varus leg axis correction in the short term (6 weeks postoperative) and long term (1 and 2 years postoperative) on standing balance characteristics, perceived pain and health-related quality of life. The main aim of this study was to estimate possible long-term effects of the HTO surgical intervention on patients’ bipedal postural control against the preoperative outcomes. We hypothesized that in the majority of patients, the HTO treatment would lead to an enhanced load proportion (percentage weight) below the foot on the affected side.

## 2. Materials and Methods

In this nonrandomized interventional study design, repeated measurements were performed on participants in a rehabilitation research laboratory after providing detailed explanations of the risks of the investigation and obtaining written informed consent from each participant. The study protocol of this time-series design was approved by the ethical committee of Martin-Luther-University Halle-Wittenberg (approval number: 2018-66). This is a follow-up study on the patients examined in the work of Brehme et al. [6].

Adult patients diagnosed with medial compartment knee OA verified by MRI were included. All included patients had moderate knee OA (Kellgren–Lawrence grade 3). The inclusion criteria were described in detail elsewhere [6]. Patients with knee ligament instability and contralateral knee OA requiring treatment were excluded, as were those with primary patellofemoral OA, bi-compartmental OA, tibial or femoral osteonecrosis and those with a history of an inflammatory rheumatic disease. Further, any patient who was unable to perform the posturography measurements owing to limited motion of the lower limb were excluded. Initially, 38 patients were screened for eligibility before HTO surgery (Figure 1). All measurements were performed at the same time of day and in a quiet room to minimize any disruptions during testing. Due to low sample size and to avoid distortion effects, female patients were excluded. Finally, 25 male patients were examined before, 6 weeks after, 1 year after and 2 years after HTO surgery.

The surgical procedure used consisted of a bi-plane medial-based open-wedge HTO, including a distal release of the superficial medial collateral ligament fibers. The aim was to shift the weight-bearing line laterally, with the post-operative mechanical axis running laterally through the tibial plateau at 62% of its entire width measured from the medial side [12]. Using standing whole leg radiographs, the amount of correction needed was determined using the Miniaci method [13]. The LOQTEQ Osteotomy Plate system (aap Implants Inc. Dover, DE, USA) was used for fixation. Only one experienced surgeon (K.B., 30 to 40 HTO surgeries yearly) performed the examination as well as surgery procedures to ensure the highest level of observation equality and to avoid any surgeon bias.

The vertical component of ground reaction force fluctuations under the left and right feet were obtained during bipedal standing (trial duration: 32 s, sampling rate: 32 Hz) using side-by-side force plates each divided into forefoot and heel components (IBS, neurodata, Vienna, Austria). This system provides a comprehensive and sufficient reference database of asymptomatic subjects (*n* = 1724) stratified by age and sex [14]. This reference database was used to select asymptomatic controls for the matched sample (selection criteria: age, sex and body height). Thus, recruitment of a control group was not necessary. The outcomes provided by the IBS system incorporating signals of four uniaxial sensors are reliable [15] and have been extensively validated [11,16,17]. The percentage weight (PW) represents pressure load under the foot at one side (sum of both sides = 100%) and contains averaged values from the forefoot and heel force plate sensors. The degree of standing steadiness was quantified in terms of the stability (ST) index with higher values indicating a more unsteady position. Standing asymmetry was quantified as the weight distribution (WD) index with higher values representing a more asymmetric standing position. Both measures represent complex outcome indices from two forefoot and two heel components. Their calculations are described elsewhere [11].

Pain intensity was derived from a single 100 mm visual analogue scale (VAS) pain measurement. The minimal clinically important difference (MCID) on VAS was shown to be 25 mm in patients with knee osteoarthritis independent of the type of treatment [18,19]. Health-related quality of life was examined using the German version of the 36-item Short Form SF-36 health survey questionnaire [20,21]. The results of the 36 questions combined into 8 different subscales were transformed into physical component summary (PCS) and mental component summary (MCS) scores. McHorney et al. [22] reported intraclass correlation coefficients from 0.65 to 0.94 across scales and patient groups. For the SF-36 health survey questionnaire, an MCID of at least 10 points was established in patients after total knee replacement [23]. Three points on the PCS or MCS scores are recommended [24].

Data are presented as mean with standard deviation (SD) and as minimum and maximum or 95% confidence interval (CI) values. The normal distribution of data was examined visually and verified using the Shapiro–Wilk test. Absolute and relative change scores were examined using the method purposed by Kaiser [25]. Percentage weight (PW) under feet before HTO surgery was compared using Student’s paired *t* test with Cohen’s d. Linear relationships between different interval scaled outcome measures were proven using Pearson’s product moment correlation. Baseline comparisons of demographic characteristics, standing steadiness and asymmetry between groups were verified using Student’s unpaired *t* tests with Cohen’s d. To assess treatment effects on PW, ST (standing steadiness), WD (standing asymmetry), perceived pain and health-related quality of life, direct comparisons were conducted using separate repeated measures analyses of variance (ANOVA) with time as the within-subject factor. Practical relevance was estimated calculating partial eta-squared (η_p_^2^) with values ≥0.01, ≥0.06 and ≥0.14 indicating small, moderate or large effects, respectively. Further, indirect within-group comparisons were conducted via pre-post changes.

## 3. Results

Orthopedic comorbidities (ankle instability, meniscal lesions, lumbar disc disease, neck implants) were present in nine (36%) patients. Twelve (48%) patients suffered from arterial hypertension. In 12 (48%) of the patients, the operated leg was their dominant (e.g., leg used to kick a ball) lower limb. The average wedge height was 10.8 mm (SD 2.2, 7.0–15.0). The magnitude of correction was not related to patients’ body mass (*r*(23) = 0.28, *p* = 0.17) or height (*r*(23) = 0.17, *p* = 0.41). The demographic characteristics of the study participants are listed in Table 1. Patients’ body mass remained at comparable levels (*p* > 0.18) during the course of the 2 year follow-up (6 weeks: 98.6 (14.3) kg, 1 year: 98.8 (14.9) kg, 2 years: 99.7 (14.9) kg).

Lower percentage weight (PW) was placed on the plates at the affected side before the HTO surgery (*t*(24) = –3.48, *p* = 0.002, *d* = –0.70). Controls’ percentage foot pressures did not differ between sides (*t*(24) = –0.41, *p* = 0.67, *d* = –0.08). Patients’ PW under the foot of the affected leg was neither associated with their body mass (*r*(23) = 0.21, *p* = 0.32) nor with their reported pain intensity (*r*(23) = –0.07, *p* = 0.74). At baseline, patients showed large (*t*(48) = 5.26, *p* < 0.001, *d* = 1.5) and moderate (*t*(48) = 2.20, *p* < 0.05, *d* = 0.62) differences from controls regarding standing steadiness and asymmetry, respectively.

Figure 2 depicts baseline comparisons of the stability index and the weight distribution index between the patients selected for HTO surgery and age and stature matched controls. Over time, the PW under the foot at the HTO side increased on average (*F*(3, 72) = 20.87, *p* < 0.001, η_p_^2^ = 0.47). On average, patients put higher PW under the foot at the operated side. In terms of standing steadiness, the average ST remained similar over the time points (*F*(3, 72) = 0.46, *p* = 0.71, η_p_^2^ = 0.02).

The time factor revealed large effects on standing asymmetry (*F*(3, 72) = 5.07, *p* = 0.003, η_p_^2^ = 0.17). Figure 3 depicts time changes according to baseline measurements.

There was a large effect of time on the reported pain intensity (Table 2). However, considerably large standard deviations were seen 1 and 2 years after surgery. The repeated measures ANOVA revealed significant effects of time on the PCS and MCS. While the reported physical health increased, mental health decreased 2 years after HTO surgery.

The improvements recorded for pain intensity were below the MCID of 25 mm at 1 and at 2 years after HTO surgery (Table 3). At 6 weeks after treatment, it is unclear whether the reduction in pain was clinically meaningful. The PCS clearly exceeded the MCID threshold of 3 at 1 and 2 years after surgery. Thus, the improvement in physical health was clinically important. By contrast, the MCS deteriorated, reaching the critical threshold at 2 years after HTO surgery.

## 4. Discussion

This prospective, longitudinal cohort study aimed at evaluating the influence of a surgical varus leg axis correction in the short term (6 weeks postoperative) and long term (1 and 2 years postoperative) on standing balance characteristics, perceived pain and health-related quality of life. Before HTO surgery, patients placed lower percentage weight on the force plates at the affected side and stood more unsteadily and asymmetrically as compared with asymptomatic controls. After surgery, the percentage weight under the foot at the HTO side increased on average, accompanied with reduced standing asymmetry one year after treatment. Standing steadiness, however, did not change over time.

In order to quantify standing steadiness, force plates are frequently used to record fluctuations of the center of foot pressure during upright standing. The stability index indicates the amount of force fluctuations among the four plates of the IBS [11] and thus how steady participants were able to stand during the trial. After HTO surgery, standing steadiness results were comparable to those before the intervention. Most probably, the postural task is not challenging enough and thus results in different solution strategies with high variability. Only 11 (44%) of the patients exceeded the values of the asymptomatic control subjects. Although patients had radiographically confirmed knee OA, the majority of them stood as steady as the controls. Further, comparisons of within-group differences (see Appendix A, Figure A2) confirm that changes occur with fairly equal distribution in both directions. However, during a more challenging postural task, Hunt et al. [5] failed to prove changes in standing steadiness 1 year after HTO surgery. In patients with knee OA, Hassan et al. [26] assessed bipedal standing steadiness with closed eyes before and after local anesthetic injections. Although reported pain was reduced considerably, postural steadiness remained unaltered. These findings suggest that the effects, either short-term or long-term, of mechanical leg axis corrections or pain reduction interventions on standing steadiness are negligible.

Weight distribution outcomes seem to be more sensitive to changes occurring due to disease progression or surgical interventions [10,27,28] (see Appendix A, Figure A1 and Figure A3). Standing asymmetry was quantified by two different outcomes: the percentage weight placed on the plates at the affected side and the weight distribution index. Both outcomes differ according to the signals they contain. The percentage weight contains ground reaction force fluctuations under one foot (right or left) and thus from two force plate sensors. The weight distribution index is a complex outcome including signals from four force plate sensors (two at each side). Consequently, the weight distribution index contains medio-lateral as well as anterior–posterior information. This complex outcome increased in patients 6 weeks after HTO surgery, indicating that the standing pattern was more altered as a result of the surgical intervention. After the first year, the patients stood more evenly distributed between the four force plates, resulting in a lowered weight distribution index. The majority of asymptomatic participants evenly distribute their weight during side-by-side standing [10,28]. Interestingly, patients differ in their strategies to load the affected side. A long-term follow-up (14.7 months) in patients who underwent total knee arthroplasty but without extension limitations revealed more percentage weight put on the operated leg [27]. Patients with present knee extension deficits placed more weight at their contralateral side. The majority of patients after total hip arthroplasty tended to load the non-operated more than the operated side [10]. Amputees, in contrast, showed lower percentage weight at their prosthetic limb [28].

Patients’ reported outcomes, especially the results for pain intensity, showed high variability 1 or 2 years after HTO surgery. This implies that long-term effects may rely on individual conditions. Two investigations [29,30] reported on 15 and 23 HTO patients 2 years postoperatively who were on average 6 and 11 years younger than our participants, respectively. Both studies found a quite comparable reduction of pain intensity. However, in the investigation with the on average younger patients, five (22%) went on to total knee arthroplasty [30]. Clinically meaningful reductions in pain intensity 1 and 2 years after HTO surgery go hand in hand with enhancements in physical but not mental health. This is in line with the reported magnitudes of change in different orthopedic conditions [31]. Independent of orthopedic condition, the greatest improvements were found for the physical dimensions, while only small to moderate improvements were observed for the mental and social dimensions. In a retrospective study on a cohort of comparable age, Bastard et al. [32] reported higher preoperative physical and mental subscores, both of which significantly improved 1 year postoperatively. Saier et al. [33] followed 64 patients 2 years after HTO surgery. They found inferior MCS scores to be influenced by psychopathological comorbidities examined preoperatively. The deterioration of mental health 2 years after HTO surgery in our participants, however, cannot be explained by the outcomes reported in this study and needs to be further investigated. Clinicians who aim to quantify standing balance in HTO patients are encouraged to use multi-plate posturographic systems. The percentage weight placed at one side is easy to interpret and can be used early after HTO surgery to judge rehabilitation progress.

This study comprises some limitations that need to be addressed: First, the data of the control group are from a reference database. Consequently, no longitudinal effects can be calculated. Second, a control group of patients who did not receive surgical treatment could not be included in this study. Third, obesity is a typical feature of HTO patients, in contrast to the selected asymptomatic participants. Therefore, a significant difference regarding body mass is inevitable. Finally, only data of male patients were analyzed in this investigation. In future studies, one may consider a time-series design with at least two pretest measurements to establish a baseline. Reducing variation existing in outcomes before the intervention can help researchers to estimate treatment effects more precisely. Moreover, patients’ baseline measurements were taken up to 82 days preoperatively, which may have impacted the results. Changes in the selected outcomes may be due to the HTO treatment. However, the measured changes caused by HTO surgery cannot be isolated from naturally occurring outcome changes. Different experiences as well as responses following the HTO surgery and post intervention phases may particularly explain the results. Further studies need to assess how many of HTO-treated patients will get a total joint replacement of the knee or hip joints and which predictive signs are to be considered.

## 5. Conclusions

The findings of this study indicate that patients who have undergone HTO surgery have reduced their pain and increased their reported physical health. These changes are clinically important 1 year after the surgical intervention. The standing steadiness of the HTO patients remains fairly unaltered over the follow-up period. Standing asymmetry, in contrast, changed considerably after HTO surgery. Patients increased their percentage weight under the foot at the operated side. The weight distribution index, in contrast, increased 6 weeks after surgery and decreased 1 and 2 years after the surgical intervention. In order to judge short-term effects, the percentage weight is recommended, whereas long-term effects can be identified either through the percentage weight or the weight distribution index.

## Figures and Tables

**Figure 1 jpm-12-01594-f001:**
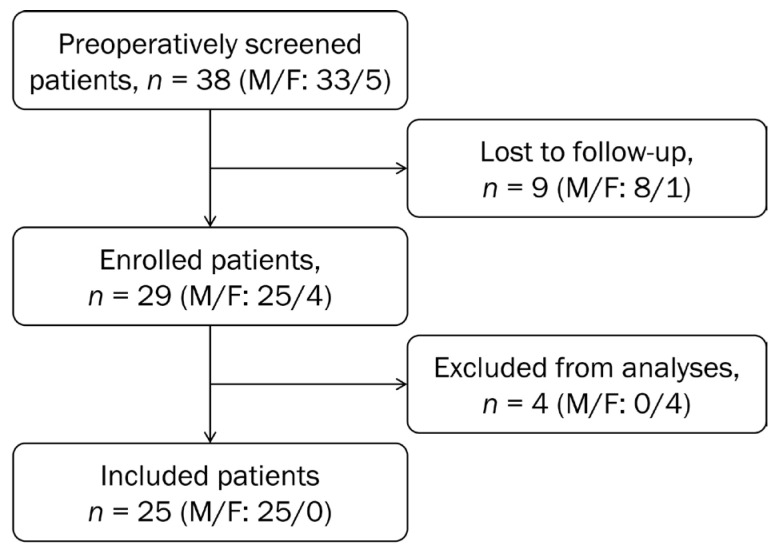
Flow chart of the study population.

**Figure 2 jpm-12-01594-f002:**
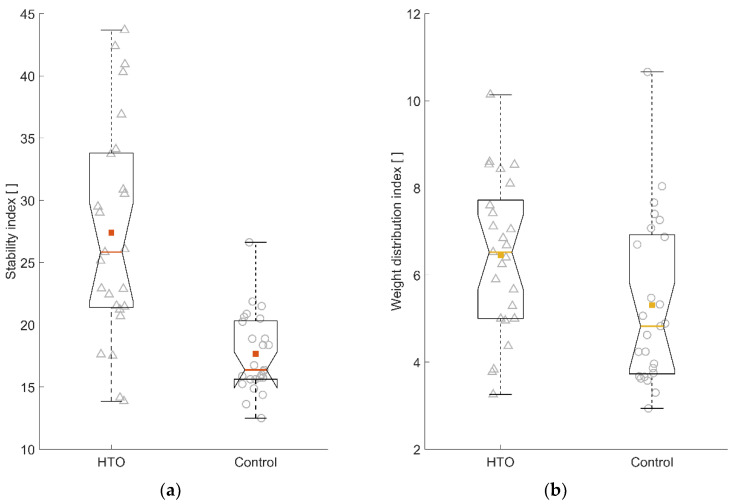
Box-whisker plots on the baseline (**a**) stability index and (**b**) weight distribution index data between the patients selected for high tibial osteotomy (HTO) surgery and matched controls with individual data points. The square symbol in the box denotes the mean of the group data. Higher values indicate greater standing unsteadiness and asymmetry, respectively. △, HTO individual data; ○, Control individual data.

**Figure 3 jpm-12-01594-f003:**
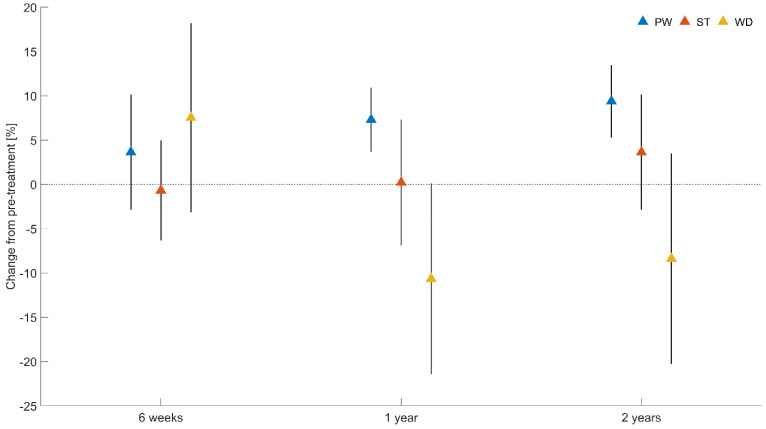
Time changes of percentage weight (PW), standing steadiness (ST) and standing asymmetry (WD), according to baseline measurements. Values are mean with 95% confidence intervals.

**Table 1 jpm-12-01594-t001:** Baseline demographic characteristics of the study participants. Values presented as mean (SD, minimum and maximum).

	HTO (*n* = 25)	Controls (*n* = 25)	*p* Value
Age [years]	56.6 (5.0, 43–62)	54.9 (7.1, 40–64)	0.35
Body mass [kg]	98.7 (14.1, 77.8–129.1)	87.1 (12.9, 65.5–129.4)	<0.01
Body height [m]	1.79 (0.06, 1.70–1.96)	1.78 (0.07, 1.66–1.95)	0.77
BMI ^1^ [kg/m^2^]	30.8 (3.7, 24.0–39.6)	27.4 (3.7, 20.5–39.5)	<0.01

^1^ Body mass index. *p* values are based on Student’s unpaired *t* tests.

**Table 2 jpm-12-01594-t002:** Baseline and follow-up results of pain intensity and health-related quality of life outcomes of the patients (*n* = 25). Values presented as mean (SD, minimum and maximum).

	Baseline	Six Weeks	One Year	Two Years	*p* Value	η_p_^2^ Value
VAS ^1^ (0–100)	49.5 (21.2)	31.8 (15.0)	20.4 (23.0)	21.0 (24.2)	<0.001	0.35
	6–100	4–56	0–72	0–86		
PCS ^2^ (0–100)	31.5 (9.0)	29.7 (8.0)	45.0 (9.9)	44.8 (12.2)	<0.001	0.56
	18–51	16–46	16–57	12–59		
MCS ^3^ (0–100)	56.8 (10.9)	55.6 (9.7)	55.1 (7.3)	51.6 (10.0)	0.01	0.14
	34–70	30–68	36–63	29–65		

^1^ Visual analogue scale: Higher values indicate greater pain intensity. ^2^ Physical component summary: Higher values indicate better physical health. ^3^ Mental component summary: Higher values indicate better mental health. *p* values are based on repeated measures ANOVA’s.

**Table 3 jpm-12-01594-t003:** Mean absolute changes of pain intensity and health-related quality of life outcomes of the patients (*n* = 25) according to baseline measurements.

	6 Weeks	1 Year	2 Years
VAS ^1^	–17.7	–29.1	–28.5
PCS ^2^	–1.7	13.6	13.3
MCS ^3^	–1.2	–1.8	–5.2

^1^ Visual analogue scale: Lower values indicate greater pain reduction. ^2^ Physical component summary of the SF-36: Higher values indicate greater improvement in physical health. ^3^ Mental component summary of the SF-36: Lower values indicate greater decline in mental health.

## Data Availability

The datasets used and analyzed during the current study are available from the corresponding author on reasonable request.

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
