# Peer review of "Standing Steadiness and Asymmetry after High Tibial Osteotomy Surgery: A 2 Year Follow-Up Study"

_jpm, 2022, doi:10.3390/jpm12101594_

Round 1

Reviewer 1 Report

It was a interesting topic about HTO. 

The adaptation and progression of limb weight bearing seems important after ipsilateral HTO.

However, the authors should modify and support the method and results parts. 

Especially, in demographic, more information of preoperative status is needed. Not only age and body mass, such factors as  previous surgical history on affected lower limb or concomitant spinal problems should be described. In addition, there was some significant uneven distribution of body weight between the HTO and control group.  Factors which can affect the weight bearing of individual lowerlimb should be controlled to produce valuable results of this study. I recommend additional propensity or matching method to make it clear and even. 

And an analysis for weight bearing of lowerlimb, the detailed descriptions of initial and postoperative alignment of lowerlimb in both groups. That might be critical point to consider the reliability of your results. Please add detalis of leg alignment and modify the demographic setting. 

In discussion and conclusion, more meaningful message of this study to readers who clinical perform HTO surgery and rehabilitation after HTO. 

Reviewer 2 Report

Although small number of patients, this paper is well written.

very interesting. I perform a significant number of HTOs, and I found this paper interesting.

Good introduction, methodology, results, discussions and conclusion.

Good and appropriate statistics.

Although the number of patients is low this paper can add to the current literature.

I would accept this paper for publication in this journal.
